# The Weather Impact on Physical Activity of 6–12 Year Old Children: A Clustered Study of the Health Oriented Pedagogical Project (HOPP)

**DOI:** 10.3390/sports8010009

**Published:** 2020-01-19

**Authors:** Iana Kharlova, Wei Hai Deng, Jostein Mamen, Asgeir Mamen, Maren Valand Fredriksen, Per Morten Fredriksen

**Affiliations:** 1School of Health Sciences, Kristiania University College, 0150 Oslo, Norwayweihaideng@gmail.com (W.H.D.); asgeir.mamen@kristiania.no (A.M.); marenvaland.fredriksen@kristiania.no (M.V.F.); 2The Norwegian Meteorological Institute, 0371 Oslo, Norway; josteinm@met.no

**Keywords:** children, physical activity level, actigraph, weather conditions, HOPP-study

## Abstract

It is commonly known that children do not engage in a sufficient amount of physical activity. Weather conditions and day length may influence physical activity of children. Little is known about the relationship between physical activity and seasons. The purpose of this study was to investigate the relationships between weather conditions and physical activity in 6–12 year old children based on hip-worn Actigraph wGT3X–BT accelerometer data. The study sample consisted of 2015 subjects aged 6–12 years from the Health Oriented Pedagogical Project (HOPP) study carried out in Horten municipality and Akershus county, Norway. Six days of sedentary and moderate-to-vigorous physical activity data was gathered in January–June and September–October, 2015, presented as daily averages. The accelerometer-monitored physical activity of children grouped within nine schools was matched with regional weather conditions and assessed with the means of linear mixed models. Increased day length was associated with decreased sedentary behavior. Warmer temperature and dry weather were associated with increased moderate-to-vigorous physical activity after adjusting for age and sex. One-hour increase in daylight resulted in a decrease of sedentary time by, on average, 2 min (95% CI = (−2.577, −0.798)). For every 5 °C increase in temperature (range: −0.95 and 15.51 °C) and dry weather, average moderate-to vigorous physical activity increased by 72 and 67 min (males and females, respectively) (*p* < 0.001). Days with precipitation had, on average, 10 fewer minutes of moderate-to-vigorous physical activity compared with days without precipitation (95% CI = (−16.704, −3.259)). Higher temperatures and dry weather led to higher physical activity levels, seeing larger increases among boys than girls. A school-based physical activity intervention program should be adjusted regarding local weather conditions in line with the present findings.

## 1. Introduction

A lot of studies have recently been carried out to inspect probable relationships between physical activity (PA) and weather conditions. A question regarding PA and its positive impact on school-aged children, as well as its promotion in schools, has been discussed recently [1,2,3,4], considering PA to be a crucial component of children’s everyday lives. It has been proposed that a consistent increase in PA could reduce obesity, heart disease, and a number of other chronic illnesses, as well as even improving cognitive performance [1,5,6,7]. It is therefore important to design productive school intervention programs to increase PA levels. That is why research has begun to investigate potential obstacles to PA and major forces causing physical inactivity.

It has been observed that children’s PA intensity shows variations depending on seasonal differences [8,9,10,11,12]. In particular, Norwegian 9-year-olds had higher PA during spring compared with fall or winter [13]. Children have been found to be more active in warm weather [14,15,16] and increasing day length [17,18], and less active with increased precipitation [9,19].

However, weather and PA measurement tools, as well as geographic locations, differed in the above-cited studies. There is, to our knowledge, a limited number of studies looking at the association between PA in children using accelerometry and local weather data, and furthermore, none that bases its analyses on hourly data. This paper is thus novel in its access to high level detailed PA and weather data. The goal of this research is, thus, to investigate PA of 6–12-year-old children given varying weather conditions, and bring forward the proposals and recommendations regarding how PA should be increased in different weather conditions for improving overall health of school-aged children.

## 2. Materials and Methods

### 2.1. Subjects and Project Study Sample

The 7-year longitudinal Health Oriented Pedagogical Project (HOPP) study was launched in 2015. It includes a population of 2816 elementary school children from nine schools with a baseline age of 6–12 years. The total sample after the process of recruitment was 2297 children, which equaled 81.6% of the population. The following cross-sectional analysis uses data obtained in 2015. Both recruitment and intervention processes are described elsewhere [20].

### 2.2. Physical Activity Assessment and Outcome Measures

Actigraph wGT3X-BT accelerometers (ActiGraph LLC, Pensacola, FL, USA) with a sampling frequency of 100 Hz and 10 s epochs were utilized to measure PA. Children received detailed directions of how to wear the accelerometers. In particular, such devices have to be worn during all hours for seven consecutive days, on the right hip, attached with an elastic band, except when swimming/showering, injured/ill, or absent from school. A recorded activity of at least eight hours per day (from 06:00 till 23:59) with a non-wear time exclusion was considered to be appropriate for the following analysis. The registered mean counts per minute (cpm) were then used to categorize the PA levels as sedentary (0–99 cpm), light (100–1999 cpm), moderate (2000–4999 cpm) and vigorous (≥5000 cpm) [21]. Moderate-to-vigorous PA was found by totaling the minutes of both moderate and vigorous intensities.

To accurately assess PA and reduce incomplete data associated with the beginning of data collection, i.e., when children received the accelerometers, only six days of measurements were used. The final study sample consisted of 2015 individual children (1020 females and 995 males, mean age 9.46 ± 1.76), approximately 88% of the HOPP sample.

### 2.3. Covariates

Meteorological data from the Norwegian Meteorological Institute were acquired from the three weather stations located closest to the schools. The first station is placed around 14 km away from the seven schools. The second and third stations are around 7 km and 5.6 km away from the eighth and ninth schools, respectively. Weather data was matched with valid hours of accelerometer data, and included hourly temperature (in degrees Celsius), precipitation intensity (in millimeters), and day length (in hours per day). Length of the day was defined as the elapsed time between sunrise and sunset. All weather measurements were averaged for each day, and were considered continuous variables. Child age and sex were used in subsequent analyses as continuous and binary variables, respectively. Children’s height, weight and waist circumference correlated with age (Pearson’s *r* = 0.85) and therefore were eliminated from the final analysis.

### 2.4. Data Analysis

To account for clustering, a linear mixed effect model was used to analyze the relationships between PA and the weather conditions, using R software version 3.5.1 [22]. The individual children were separated by the nine recruitment schools during four seasons with contrasting weather conditions that are associated with their own explanatory variables. Variations in sedentary behavior and moderate-to-vigorous PA (MVPA) between schools is shown in Figure 1.

To handle this by-school variation that could not be explained by the covariates, inclusion of a random effect for the school was necessary. To estimate the outcomes, sedentary time and MVPA, the following fixed effects were used—weather conditions (average temperatures, precipitation and daylight hours) as well as sex and age. The analysis started with a model including only a fixed effect for the intercept. Then, it was adjusted for the individual level covariates, i.e., sex and age, and weather-related variables. To avoid bias for the random component given by maximum likelihood estimator, restricted maximum likelihood estimation (REML) procedure was applied to find parameter estimates [23]. As a model selection criterion, the Akaike information criterion (AIC) was applied. Conditional residuals were examined to diagnose whether the model fitted the data well. To detect multicollinearity in adjusted models, the variance inflation factor (VIF) was calculated. An accepted cutoff for significance was set to 0.05.

### 2.5. Ethical Considerations

The research methods applied in this study adhere to the ethical principles governed by the World Medical Association’s Declaration of Helsinki and subsequent revisions. The research protocol (ref.no.:2014/2064/REK) was approved by The Regional Committee for Medical Research Ethics (REK). The research is registered in Clinical Trials (Identifier: NCT02495714). Parents or legal guardians of children gave informed consent for their children to be research subjects. Personal data are not published.

## 3. Results

### 3.1. General Findings

This research revealed that males were more physically active than females at all ages (Figure 1). Furthermore, measured average daily MVPA steadily decreased with age (Figure 2) among Norwegian children aged 6–12 years (Figure 2).

### 3.2. Sedentary Behavior

Table A1 shows the results of linear mixed models of average sedentary time, with a random intercept for each school.

The crude models showed statistically significant associations between all variables and sedentary time, except temperature (*p* = 0.085). Increased daylight hours were associated with decreased sedentary activity, and increased precipitation with increased sedentary time. Additionally, sedentary time went up with each one-year increase in age of a child. Male children were observed to have less sedentary time. Residual inspection showed patterns of homoscedasticity and linearity. All the covariates showed low correlation with each other (Pearson’s *r* = −0.12 to 0.07). Daylight hours and temperature were moderately correlated (Pearson’s *r* = 0.63).

Pairwise interactions between weather variables were not statistically significant. Including them into models led to moderate and high multicollinearity (VIF range: 6.78 to 10.27). According to the lowest REML-based AIC, the model containing age, sex and daylight was chosen as the final model for the sedentary time analysis. However, after adjustment for precipitation and temperature, the estimated fixed effect of these covariates was not statistically significant, and their effect on sedentary activity stayed unclear. Variability of the intercept across schools equaled 23.2. Younger children spent less time staying sedentary compared with older children. For every additional year in age, expected daily sedentary time increased by an average of 4.5 min. In addition, sedentary activity differed significantly across the sexes. Male children had lower total sedentary time than females. Compared with females, male children, on average, had a sedentary time of 4 min less. Less hours of daylight were associated with increased sedentary activity. On average, each hour decrease in daylight resulted in a 2 min decrease in SED. The intercept equaled 226 min, and represented the average of sedentary activity for nine schools which is also shown in in Figure 1a. Visual inspection of the residual plots did not indicate any patterns in the spread. The normal probability plot of the residuals followed a straight line. The effects of the estimated coefficients are shown in Figure 3a.

A mixed effects model using random effects, representing school-to-school deviation from the population average, allows subject-specific inference and better understanding of association between fixed effects and the outcome. Assuming a model with fixed slopes and random intercepts, sedentary time of a child can be found as follows:(1)$$SED_{ij}={\hat{\tau }}_{00}+{\hat{\tau }}_{0j}+{\hat{\tau }}_{1}\times age+{\hat{\tau }}_{2}\times sex+{\hat{\tau }}_{3}\times daylight+\varepsilon_{ij}.$$

Equation (1) represents a regression or prediction of sedentary time as a function of age, sex and daylight hours, where $SED_{ij}\;$—sedentary time of child *i* in school *j*; ${\hat{\tau }}_{00}$—estimated fixed intercept; ${\hat{\tau }}_{0j}$—random effect for the intercept of school *j*; ${\hat{\tau }}_{1}$, ${\hat{\tau }}_{2}$, ${\hat{\tau }}_{3}$—fixed-effects coefficient estimates; $\varepsilon_{ij}—$random error within the *i*th child in the *j*th school.

Adjusting temperature in a model consisting of sex and age, temperature showed a significant relationship with SED (*p* = 0.002). The difference in AIC between this model and model (1) was little (17445.37, 17449.29), and the data could be described by the former model, as well. This allows the conclusion that both models could be comparable. However, visual analysis showed that daylight hours had a larger impact on SED than temperature did (based on the steeper slope coefficient).

### 3.3. MVPA

Table A2 shows the results of linear mixed models of average MVPA, with a random intercept for each school.

The crude models showed statistically significant associations between all variables and MVPA. A one-year increase in age was associated with declined MVPA. Male children had higher MVPA compared with females. Increase in both temperature and daylight hours was expected to give raise to MVPA. Additionally, increased amount of precipitation was associated with reduction in MVPA. Analysis of residual plots showed homogeneity and normal distribution.

The minimum AIC value was obtained for a model including age, sex, temperature and precipitation (the results are presented in Table A2). After adjusting for daylight hours, the latter did not contribute to the changes in MVPA. On average, MVPA decreased by around 4 min per day with each one-year increase in age. Male children appeared to have higher MVPA than females. Time spent in MVPA for female children was, on average, lower by about 5 min, when compared to males. The estimated fixed effect of precipitation on MVPA was negative (−10.207), suggesting that an increase in precipitation intensity by 1 mm was associated with a lower predicted MVPA, after adjusting for the effects of other covariates (i.e., age, sex, and temperature). In addition, children were predicted to have an average daily MVPA about 1 min higher with a one-degree temperature increase after adjusting for the effects of other covariates in the model. The estimated among-school variance equaled 4.976. No significant interactions were found. The estimated fixed effects are shown in Figure 3b.

A regression model predicting MVPA of a child with the same slope in each of the schools, and varying intercepts is given by:(2)$$MVPA_{ij}={\hat{\tau }}_{00}+{\hat{\tau }}_{0j}+{\hat{\tau }}_{1}\times age+{\hat{\tau }}_{2}\times sex+{\hat{\tau }}_{3}\times temperature+{\hat{\tau }}_{4}\times precipitation+\varepsilon_{ij},$$
where $MVPA_{ij}$—MVPA of child *i* in school *j*;$\;{\hat{\tau }}_{00}$—estimated fixed intercept; ${\hat{\tau }}_{0j}$—random effect for the intercept of school *j*;${\hat{\tau }}_{1}$, ${\hat{\tau }}_{2}$, ${\hat{\tau }}_{3}$, ${\hat{\tau }}_{4}$—fixed-effects coefficient estimates; $\varepsilon_{ij}$—random error within the *i*th child in the *j*th school.

### 3.4. Seasonal Variations in PA

Figure 4 shows dot plots for the estimated school-specific random effects for the fully adjusted models presented in Table A1 and Table A2, arranged by increasing values of the random intercepts. Values lying under the horizontal line imply low average PA, and values above the horizontal line specify high average PA after adjusting for the other covariates. Clustering within the schools is noticeable, and the coefficients approximately follow the line. The discrepancy from the line (the overall fixed intercept) is indicated by the school-level standard deviation (2.23 and 4.82 for MVPA and sedentary activity, respectively). Some variations in MVPA and SED coming from the schools were found after adjustment for weather-related covariates and individual level explanatory variables.

Children in schools 8 and 6 (tested in June and April), on average, seemed to be more active given the observed weather conditions (Figure 4). Children in schools 3 and 5, (tested in March and April) had lower than average MVPA and higher average sedentary activity adjusted for the effect of other covariates (sex, age, and weather condition). Children in school 4 (tested in March) had average intensities of MVPA and sedentary activity given the observed weather conditions and individual level covariates. MVPA and sedentary time correlated positively in schools 9 and 1 (tested in September and January) with regards to the local weather.

Children in schools 5 and 6 were examined during similar weather conditions (average temperature of 9 °C, 16 h of daylight and light precipitation intensity), but were ranked oppositely with respect to PA according to Figure 4. Children in schools 7 and 8 were also tested under identical weather conditions (average temperature of 10 °C, 17 h of daylight and light precipitation intensity) and had the same average sedentary time, but different MVPA.

## 4. Discussion

### 4.1. Summary

Sedentary behavior was significantly affected by day length in a linear relationship. A one hour decrease in daylight was associated with two more minutes of sedentary time per day, agreeing with previous research on school-aged children [15,17,18]. However, some previous studies using pedometers [24] did not show any significant associations between day length and PA. The probable reason is demographic or environmental differences (accessibility of facilities).

Higher temperatures were associated with one more minute of MVPA, adjusting for individual-level covariates. This implies that on average, MVPA of 10-year old male children could increase by almost 71.5 min per day with a 5-degree increase in temperature assuming dry weather. Also, MVPA of 10-year old female children could, on average, increase by around 67 min per day with a 5-degree increase in temperature and dry conditions (in a range of temperatures between −0.95 and 15.51 °C). The association between precipitation and PA was inverse, suggesting a decline in MVPA by around 10 min per day with a 1 mm increase in precipitation. Also, the effect of precipitation on children’s accelerometer-measured PA was discussed in some of the past studies, and is congruent with the current results [14,19,25].

Recent studies redouble the linear relationship between temperature and PA found in this paper [14,15]. However, a curvilinear relationship between temperature and PA between 20 and 33 °C was pointed out, suggesting the highest PA levels occurred at temperatures above 22 °C [14,16]. In the present analysis, temperature varied in the range between −0.95 and 15.51 °C, and thus, an indicated linear relationship appeared to agree with the above-mentioned research.

Another finding was that average daily measured MVPA decreased with children’s age (Figure 2b). This result is consistent with previous studies of Norwegian children [26,27], as well as other research carried out worldwide [28,29,30]. This pattern can probably be explained with increases in mass of older children leading to increases in body inertia. Besides this, the current analysis indicated significant sex differences in PA, supporting previous studies [31,32,33,34].

### 4.2. Variations in PA between Seasons

Previous studies have found that children are more active during summer compared with other seasons [8,9,11,12,35,36]. Norwegian 9-year-olds were generally found to be more active in spring than in winter or fall, though no seasonal variations were found among 15-year-olds [13]. The small role that seasons play on MVPA can be linked to lower SED in the summer [10].

The design of this study required inclusion of a random effect that was estimated in conjunction with the clustering structure. The nine schools were considered as a small sample from a sample space. Among-schools variation was regarded to be a nuisance variable, decreasing the accuracy of the estimated fixed effects. However, this research allows inference to be made about seasonal variations in PA as children were examined during various seasons.

No distinct patterns were observed in the association between seasonality and MVPA or SED according to Figure 4. A possible reason may be that children were not examined during the summer holidays (i.e, July and August), which could probably prevent the adequate assessment of PA from season to season. Another reason is the geographical location of the schools. The nine schools are settled in similar weather conditions, that is, seven are placed along the Oslo fjord, and two of them are located east and west of Oslo. It might not be expected to detect any substantial seasonal variations in roughly the same surroundings. For this reason, further research might be required in the other parts of Norway, e.g., the extreme western coast and northernmost regions characterized by heavy precipitation intensity and severe low temperatures.

### 4.3. Strengths and Limitations

The main strength of this research is that it can be approximated on a much larger population of school-aged children due to random effects approach used. It also provides fixed effects to estimate child-specific PA levels depending on weather conditions. Validity is further strengthened by the use of triaxial accelerometers for PA assessment [12,37].

Four major limitations in this research should be addressed in future studies. First are the cut points used to categorize PA intensities, which is a subject of deep discussion [38,39,40]. The validity of various cut points should be carefully examined for different populations. Furthermore, the study was restricted by the location of the HOPP project, characterized by a relatively homogeneous region with mostly similar weather conditions [20]. This limitation did not allow to find the association between PA and unusual weather patterns for the HOPP-area. The nearest weather observation stations that provided weather-related data are located far away from the schools that are engaged in the HOPP project: 14 km, 7 km, and 5.6 km away. Such a significant distance between the schools and weather stations might bring inaccuracy to the actual weather conditions that were experienced by children. This issue could be controlled by using weather data obtained from remote weather stations located in schools. However, such data would not be validated. Lastly, the length of the day was defined in this study as the time period between sunrise and sunset, excluding dusk during which children could also perform minor PA.

## 5. Conclusions

This study showed that the day length affects average SED of children, and precipitation intensity and temperature influence average MVPA. The importance and value of these findings is that they give a clearer insight into when and how to set up interventions at schools to monitor progress in PA of children. Particularly, this suggests that PA should be increased during cold and wet weather conditions. No variation in PA was found in regards to seasons. This implies that the short-term changes in the weather could influence PA more than long-term periodic variations in weather, such as those caused by seasons. Further analysis is required in different parts of Norway to gain a broader insight into whether geographic factors affecting local weather patterns influence PA. Additional weather related variables could be used in future research, such as humidity, wind speed atmospheric pressure, and solar radiation.

## Figures and Tables

**Figure 1 sports-08-00009-f001:**
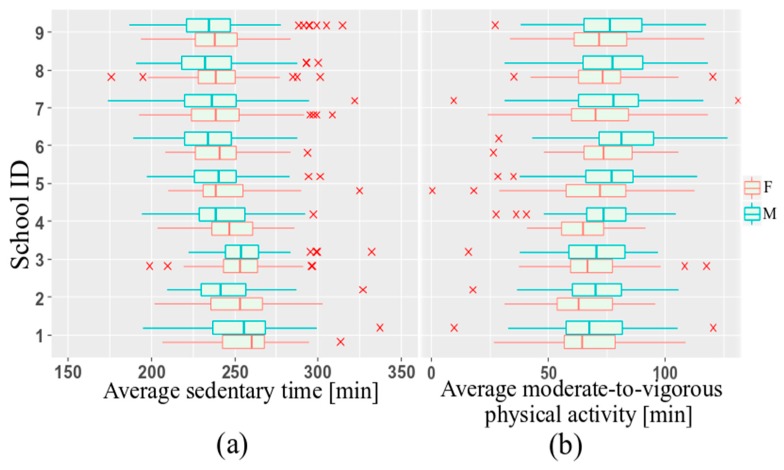
Average SED (**a**) and MVPA (**b**) per day, grouped by sex.

**Figure 2 sports-08-00009-f002:**
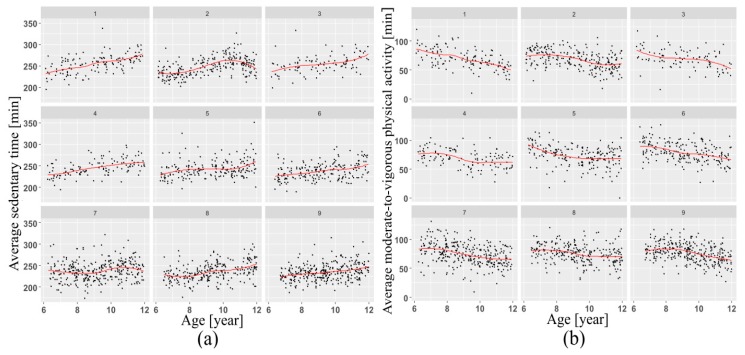
Change in SED (**a**) and MVA (**b**), grouped by age and schools with a loess smoother with a span of 0.5 superimposed, showing increasing patterns in sedentary activity and decreasing MVPA.

**Figure 3 sports-08-00009-f003:**
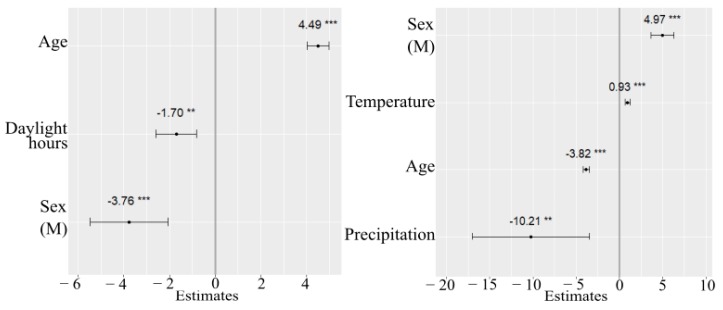
Fitted values of fixed effects of SED (**a**) and MVPA (**b**) with 95% CI for adjusted models. *** *p* < 0.01, ** *p* = 0.04.

**Figure 4 sports-08-00009-f004:**
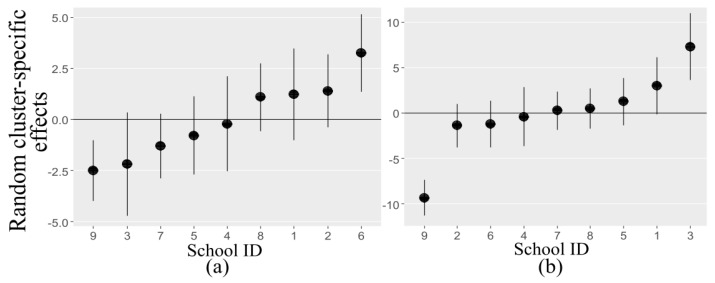
Estimated variability of the random effects of MVPA (**a**) and SED (**b**) associated with each school, with 95% CI.

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
