# Peer review of "The Weather Impact on Physical Activity of 6–12 Year Old Children: A Clustered Study of the Health Oriented Pedagogical Project (HOPP)"

_sports, 2020, doi:10.3390/sports8010009_

Round 1
Reviewer 1 Report
This is an important topic that adds to the database about factors affecting childhood obesity and possible motivations for children (and their parents) getting enough daily activity. Perhaps it would be helpful to comment on whether or not children in those schools also participated in regular swimming classes through the school or otherwise, as they would probably not be wearing the study device for that activity and so it might not be measured. It would also be helpful to know if the children (or parents) kept some kind of written log in addition to the numbers collected by the study device.
In addition, perhaps the graphics could include some more visual trends--months of the year (and/or daylight hours) on the x axis and level of activity on the y axis. Overlay with precipitation trends. it would be nice to capture the numerical descriptions with a more visual display of the data.
Author Response
Thank you for sharing your feedback. Please see the following responses to your suggestions.
Response 1.
There are swimming classes in Norwegian schools, but they are not mandatory, so children are free to choose whether they would like to attend or not. Also, the device that was used (Actigraph wGT3X-BT) is not water-resistant, and should never be placed in water, otherwise it will be destroyed, and all information stored on it will be lost. That is why it must be taken off while swimming or showering.
Response 2.
Referring to your second suggestion, please see section 2.5 about ethical considerations. No one has access to the collected data. It might be published after the study is finished only upon the study lead's approval.
Response 3.
Please see Figure 3. The activity level is already depicted on the y-axis. Your suggestion regarding showing the weather condition trends graphically is quite reasonable and would give more insight. However, please note that the weather data was acquired during one week of each month when children were tested. This means that this data is not time series data (which usually are continuously measured during a long time period). This is the reason why it would be quite difficult to plot a nice and smooth line representing either precipitation, temperature or daylight hours. If the weather data was in the form of a time series, I would without a doubt superimpose it on Figure 3.
Reviewer 2 Report
Dear Authors
Authors performed an original research about the relationship among weather conditions, day length, and physical activity of children. Sample size is adequate and recording techniques are clear. In addition, although this manuscript has interesting merits and accuracy, some minor issues can be addressed to improve the quality and comprehensibility. In other words, this manuscript need to some modifications and supplements as follows.
Three major comments in your research as follows;
How do you categorize the various intensities of physical activity? Perhaps the weather conditions felt by subjects will vary because the distances between the students' school and the weather stations are all different. How will you control and interpret it? What information would you like to share with your readers? The conclusion is incomplete.
Three minor comments in your research as follows;
Line 64: Is there any special reason to select children between the ages of 6-12? Line 53-57: If you set your study as middle school or high school students, would you expect different results? Why do all the tables show SE rather than SD?
Best wishes,
Author Response
Thank you for sharing your feedback. Please see the following responses to your comments.
Response 1 (three major comments).
The categorization of physical activity intensities is described in section 2.2. Briefly, it is based on counts per minute (cpm). This division is quite subjective though. There is no general rule or any guidelines on which the amount of cpm should be considered as sedentary, vigorous, etc. The only thing that should be kept in mind is the age group of the population, as physical activity will differ (and therefore so will cpm) with age. In this research, such a division is considered to be kind of a standard for Norwegian children.
The meteorological stations that provided the weather data are located quite far away from the schools. However, they are the nearest stations available in this particular geographical location. The weather conditions could probably vary a bit in that area where the schools are located. Nevertheless, the location of any particular meteorological station is not random at all, chosen as the area around it stays nearly the same for several kilometers in terms of geography, latitude, etc. This means that in the context of meteorology there is nothing wrong with such weather data, and it could be considered correct, even though we accept that some minor changes in weather remain possible away from the stations. This information is very specific and is out of the scope of this research. The only thing that could perhaps be used to control such small weather differences is to use data from some remote weather stations that could be located in schools. However, such data would not be validated. I will address this issue very briefly in conclusion.
Response 2 (three minor comments).
Children start school when they are 6 years old and finish secondary school when they are 12. The HOPP study is a 7-year longitudinal study that includes children of ages 6 - 12. The baseline data, i.e. obtained in the year of 2015 (used in this analysis), containes all the educational years and children of all ages (6 - 12). The number of children is reduced with each subsequent educational year. The aim of the study is to follow up children and monitor changes in their physical activity throughout the 7-year period. Please refer to Fredriksen et al. The Health Oriented Pedagogical Project (HOPP) - a Controlled Longitudinal School-Based Physical Activity Intervention Program (reference 20) for the detailed HOPP study description.
We would surely expect different results if the study was conducted on high-school students or generally on older students. The potential reason could be perhaps both greater body mass and increased sedentariness of students, as they usually have more classes and homework when they are in high school.
SD and SE are two totally different things, and they are usually misinterpreted. SD is used when describing the population, i.e. it shows a variation on both sides of the mean. SE, on the other hand, shows the accuracy with which the coefficient was estimated. Since the tables contain the estimates of the regression, SE is correct to use. More generally, SD is used for descriptive purposes and SE - for inferential purposes.
Reviewer 3 Report
The manuscript, "The Weather Impact on Physical Activity of 6 – 12 Year Old Children: A Clustered Study of The Health Oriented Pedagogical Project (HOPP)" presents a original approach to the investigation of physical activity level (PA) in children. Authors discussed influence of varying weather conditions on PA, and presented forward the proposals and recommendations regarding how PA should be increased in different weather conditions for improving overall health of school-aged children. The characterization of the methods and the results achieved are compelling and indicate that the short-term changes in the weather could influence PA more than long-term periodic variations in weather, such as those caused by seasons. The importance and value of this study is that they give a clearer insight into when and how to set up interventions at schools to monitor progress in PA of children.
Finally the authors suggest that further analysis is required in different geographic regions to gain a broader insight into whether geographic factors affecting local weather patterns influence PA.
The manuscript is well written and interesting to read, however I see the following major issues that should be resolved before publishing this paper:
1) Data presentation
Table 1 (p4,l122) selected data from these table are duplicate on figure 2 (p5, l146). Data presented in graphical form are clear and easy to interpretation for readers. Likewise chart is more appropriate in regards to visualization the main results from values presented in Table 2 (p5,l167). Finally both table 1 and 2 should be placed in the supplementary data section.
Figures: 1, 3 and 4 as well their descriptions should be placed in the results section.
2) Specific comments:
Description of abbreviations "CI – 95% confidence interval, SE – standard error" (Table 1 and 2) should be placed at the bottom of the tables.
3) Comments to future research: in further research I suggest to consider additional environmental factors affecting exercise performance in children such as: humidity, wind speed, barometric pressure, solar radiation.
Author Response
Thank you for sharing your feedback. Please see the following responses to your comments.
1) Data presentation
Both Table 1 & Table 2 are now in Appendix A.
It makes no sense to place Fig 1 in the results section because this figure explains and shows exactly why I used linear mixed model and not just a linear model. That is why the figure is currently placed in the data analysis section because that is where I write about the method that was used to analyze the data. In this particular research, the inclusion of a random effect associated with schools was necessary and an absolute must. Fig 1 shows variations in PA among schools. Such a variation is best modeled by a mixed model. Without this figure in the data analysis section, it is hard to explain why I used mixed a model. This figure explains the choice of the data analysis method. Fig 1 shows the model choice, it does not show anything special that could be addressed in the discussion section. However, if you insist, I will move this figure to the results section.
Figures 1, 3 & 4 are now in Results.
2) Specific comments
These were placed at the bottom.
3) Comments to future research
A brief sentence regarding your suggestion was added to the manuscript.